

# The influence of the Atlantic Multidecadal Variability on Storm Babet-like events

Vikki Thompson[1], Sjoukje Philip[1], Izidine Pinto[1], Sarah Kew[1]

5    [1] Royal Netherlands Meteorological Institute (KNMI), De Bilt, Netherlands.

*Correspondence to*: Vikki Thompson (vikki.thompson@knmi.nl)



**Abstract.** In October 2023 Storm Babet led to extreme flooding and strong winds across the United Kingdom. We use atmospheric flow analogues to assess multidecadal variability in events similar to Storm Babet. We show that comparing analogues for timeslices results are highly sensitive to the chosen periods, thus we instead assess analogue trends through time. We identify a possible link between Storm Babet-like events and the Atlantic Multidecadal Variability (AMV), supporting the hypothesis that positive AMV leads to stormier western European weather. Events similar to Storm Babet are 7.5 times more likely during positive AMV. The method presented could be developed for use in the attribution of extreme weather events, allowing identification of possible causes of changes in the similarity of analogues to an extreme event through time. Increasing our understanding of the causes of extreme weather events can allow us to better predict future changes in such events, allowing society to prepare and adapt for the future.

## 1 Introduction

In October 2023 Storm Babet led to severe and widespread weather impacts for many parts of the United Kingdon (UK). The storm brought strong winds and exceptional rainfall to Northern Britain, with Eastern Scotland and north-east England most heavily impacted (Kendon, 2023). Such events lead us to question the roles of anthropogenic forcings and internal variability. Attribution studies of Storm Babet have suggested that climate change played a role in the wind intensity and rainfall amounts (Clarke et al., 2024; Ginesta-Fernandez & Faranda, 2023). Ginesta-Fernandez & Faranda (2023) note that the Atlantic Multidecadal Variability (AMV) and Pacific Decadal Oscillation (PDO) may have partly influenced the event. Applying multiple methods to assess an event is useful, as it can provide greater confidence in the findings. In this study we apply a new method using atmospheric analogues to explore the role of the AMV in modulating multidecadal trends in frequency of stormtims similar to Storm Babet.

The AMV is a mode of observed multidecadal climate variability, with alternating warm and cool phases of North Atlantic sea surface temperature (Knight et al., 2006). Also referred to as the Atlantic Multidecadal Oscillation, given doubts about its oscillatory nature due to the short observational record available we use the term AMV (Hodson et al., 2022). The AMV is likely driven by multidecadal variability in the Atlantic Meridional Overturning Circulation (AMOC) (Zhang et al., 2019). Many regions show climatic variability linked to the phase of the AMV, for example through its influence on the location of the Atlantic Intertropical Convergence Zone it can affect Amazonian rainfall (Folland et al., 2001), and a positive AMV phase can lead to an enhanced Western African Monsoon (Mohino et al., 2024). The AMV can be linked to European storminess through its influence on the North Atlantic Oscillation (NAO) (Ruggieri et al., 2021; Msadek et al. 2011; Davini et al. 2015; Peings and Magnusdottir 2014, 2016). The AMV influences the spatial pattern of the NAO, influencing storm tracks across the North Atlantic (Börgel et al., 2020). Positive NAO is linked to stormier conditions in the UK and western



Europe (Feser et al., 2015; Wanner et al., 2001; Pinto and Raible, 2012) – hence it could be inferred that positive AMV may lead to more events similar to Storm Babet. In this study we will explore this hypothesis further.

40       Climate analogues – events with similar atmospheric circulation patterns – are useful in assessing possible changes in the dynamics of specific extreme events (Yiou et al., 2012; Faranda et al., 2022). Searching for analogues within different time periods we can assess the changes between periods – and as we know climate has changed between the periods it is often inferred that the change in analogues is the effect of climate change. However, when only comparing analogues from different time periods it is complex to determine the causes of any apparent trend or change in the pattern. The influence of 45  other drivers, for example multidecadal climate indices, could also lead to the differences between time periods. We present an alternative method, investigating the temporal variability in analogues – using Storm Babet as a case study. We begin by showing how characteristics of days with similar atmospheric circulation patterns to Babet, the event analogues, show consistent rainfall and wind patterns. Then we assess how the analogues vary between different time periods, showing decadal variability in the quality of the analogues, investigating the relationship with phases of AMV.

## 2. Data and Methods

### 2.1 Data

As a proxy for observations we use the fifth generation ECMWF atmospheric reanalysis dataset, ERA-5, from 1950 to 2023 (Bell et al., 2021). This reanalysis dataset provides spatially complete gridded fields of climate variables by combining observational records with data from forecasting models, with data assimilation systems filling gaps where direct 55  observations are unavailable or unreliable. Although the surface variables over the study region are generally reliable in direct observations, we use the ERA-5 reanalysis data to maintain spatial, temporal, and physical consistency between the variables assessed. We use daily sea level pressure, daily rainfall total, and daily mean wind fields.

       Sea level pressure data is used to define the atmospheric circulation pattern. Although 500 hPa geopotential height is often used to define the circulation patterns of extreme events we find that for this particular event sea level pressure 60  provides analogues with surface impacts more similar to the reference date (Figure A1). Sea level pressure has been shown to be capable of tracking extratropical cyclones, such as Storm Babet (Walker et al., 2020). It is used for similar applications using analogue methods, as it is closely constrained from station observations (Faranda et al., 2022).

       For Storm Babet we take 20th October 2023 as the reference date. We use the region 40 °N to 65 °N, 20 °W to 20 °E to identify atmospheric analogues of the event. Two regions of surface impacts are investigated, chosen for consistent 65  surface impacts in analogues and known impacts from the event (Kendon, 2023). Rainfall is assessed in a region of North East Scotland, 55.5 °N to 57.5 °N 2 °W to 4 °W (box in Figure 1e), and wind in a region of the North Sea, 55 °N to 59 °N 0 °E to 8 °E (box in Figure 1f).

       We use timeseries of AMV and global mean surface temperature (GMST) derived from HadISST data (Kennedy et al., 2019; Morice et al., 2021) retrieved from the KNMI Climate Explorer. Rather than using the AMV time series directly,



we use a timeseries detrended by regressing against GMST (van Oldenborgh et al., 2009, Climate Explorer). This is done to remove the effect of climate change in the AMV.

## 2.2 Event Analogues and Similarity

We assess the similarity of the reference event to every other boreal autumn (September-October-November) day, from 1950 to 2023. We determine similarity by calculating the Euclidean distance of the sea level pressure field over the event domain

(Faranda et al., 2022). Euclidean distance is used, rather than alternative methods such as spatial rank correlation, as it favours larger structures and assess overall proximity of maps in terms of mean state as well as the spatial pattern (Yiou et al., 2012). We transform the Euclidean distance values (where a smaller value indicates a more similar field) into measure of similarity (where a larger value indicates greater similarity) using the equation:

$$S = 1 - ED/ED_{max}$$

Where $S$ is the new Similarity measure, $ED$ is the Euclidean Distance of the particular date to the reference event as observed on the 20th October 2023, and $ED_{max}$ is the maximum Euclidean distance over the whole period assessed. Figure A2a shows the $S$ for each date compared with the event. The analogue sets are the days with the most similar sea level pressure field to the reference event– the greatest $S$. We assess analogue sets across two different time periods. For the satellite period we compare the timeslices 1979-2002 with 2001-2023, and for the ERA-5 period we compare the timeslices

1950-1981 with 1992-2023. For each period we take the best 1% of analogue days of the reference date – 20 days for the 22-year long satellite period timeslices and 27 days for the 30-year long ERA-5 period timeslices. To avoid sampling an analogue multiple times we impose a gap of 5 days between the analogues, and the reference event itself is excluded.

We choose to use timeslice of different lengths for the satellite period compared to the ERA-5 period (22-years and 30-years respectively). For the ERA-5 period, 1950-2023, we take 30-year timeslices, consistent with the length commonly

used for climatology, long enough to sample multiple phases of decadal variability modes. For the satellite period, 1979-2023, we use shorter 22-year timeslices to reduce overlap between the two timeslices and to be consistent with the methods rapid attribution framework: ClimaMeter (Faranda et al. preprint). We tested the influence of the longer timeslices for the ERA-5 period, by repeating using 22-year timeslices (1950-1971 and 2001-2023), finding similar results to Fig.2 (not shown).

**Multidecadal variability in Similarity**

The timeseries of annual maximum similarity (Sx) from 1950 to 2022 is selected from $S$, and the decadal variability in this timeseries is investigated. Using different metrics to create an annual similarity timeseries were explored, including annual 95th percentile and annual mean, they were found to give consistent end results, annual maximum was chosen to allow the return periods to be later calculated (Fig.A2). As AMV is a multidecadal mode of variability, a 10-year rolling mean is

applied to the Sx, AMV, and GMST timeseries. From here on, Sx, AMV, and GMST refer to the 10-year rolling mean timeseries.



Linear multiple regression is used to identify the potential relative influence of AMV and GMST on the chance of analogues occurring. Sx can be predicted using AMV and GMST through the equation:

$$Sx = \alpha\, AMV + \beta\, GMST + c$$

Where α is the coefficient for AMV, β is the coefficient for GMST, and c is a constant. The coefficients can be used to interpret which has greater influence.

We find AMV is affecting Sx (see Results), and use extreme value theory to assess the change in return period of Sx with different AMV phase. We use the generalised extreme value distribution, as we are using annual (block) maxima (Coles, 2001; Philip et al., 2020). We assume that the trend shifts with AMV, this is factored into the GEV by allowing the 110 location parameter to be linearly dependent on the AMV. The distribution is then evaluated at two values of AMV: +0.25 °C (positive phase) and –0.25 °C (negative phase). This is similar to how GEV can be used to assess a factual and counterfactual world, e.g. with and without climate change (van Oldenburgh et al., 2021; Philip et al., 2020).

## 3. Results

### 3.1 Analogues of Storm Babet

To show the value of assessing atmospheric analogues of Storm Babet, firstly we identify the closest analogues of the sea level pressure, between 1950 and 2022 (Fig.1). The observed event shows low pressure centred on the English Channel (Fig.1a), the composite of the top 20 analogues shows a similar pattern, though the low pressure is not as deep (Fig.1d). The event led to heavy rainfall over North-Eastern Scotland (Fig.1b), this is weakly, but significantly, detected in the analogues composite (Fig.1e). The heavy rainfall across Wales and England is not apparent in the analogues. The analogues capture a 120 wind pattern similar to the event, with significant and high wind speed between Scotland and Norway (Fig.1c, f).




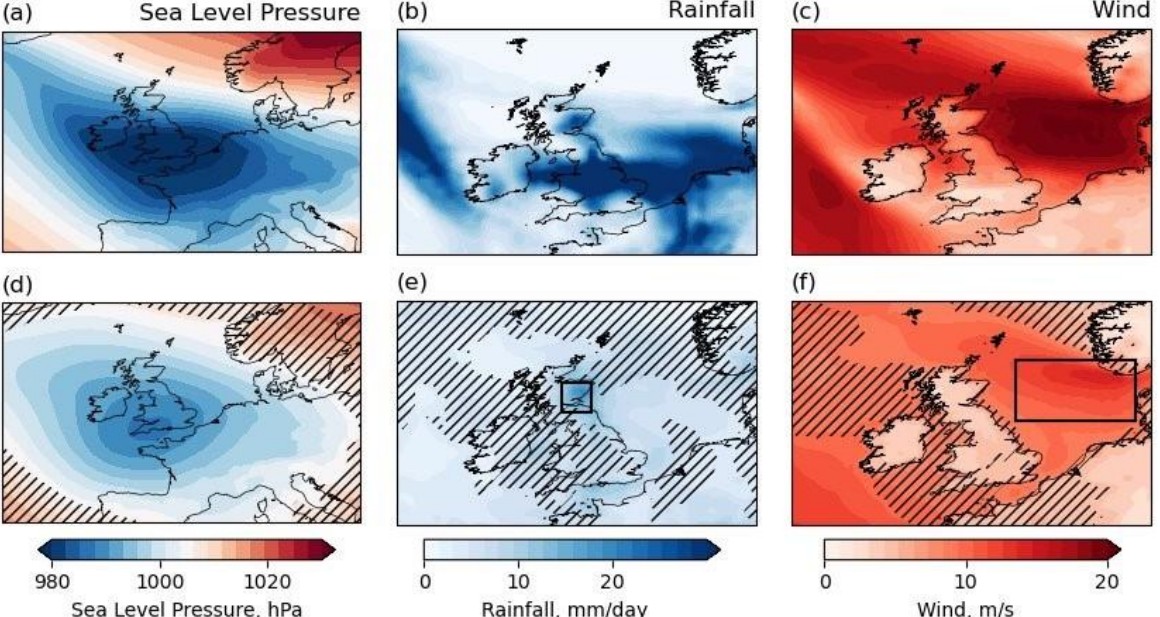

**Figure 1: Storm Babet and its top analogues. (a-c)** The meteorological situation on 20th October 2023, showing **(a)** sea level pressure
(hPa), **(b)** daily rainfall (mm/day), and **(c)** daily mean wind field (m/s). **(d-f)** Composites of the 20 closest analogues from 1950 to 2022,
based on sea level pressure, for **(d)** sea level pressure (hPa), **(e)** daily rainfall (mm/day), and **(f)** daily mean wind field (m/s). All data from
ERA-5. In **(d-f)** hashing indicates regions where the pattern is insignificant (mean anomaly < standard deviation). The regions indicated in
(e), North East Scotland, and (f), North Sea, are assessed further in Fig.5.

We compare composites of the closest analogues for two pairs of time periods (Fig.2). A pair representing the
satellite period, 1979-2002 and 2001-2023 (Fig.2a-c), and a pair representing the longer ERA-5 period, 1950-1981 and 1992-
2023 (Fig2.d-f). Assessing the change between past to present composites we show large variations depending on which
time periods are used. Indeed, opposite trends are shown over the UK, where the greatest impacts occurred. This shows that
great care must be taken when interpreting the cause of differences between time periods.



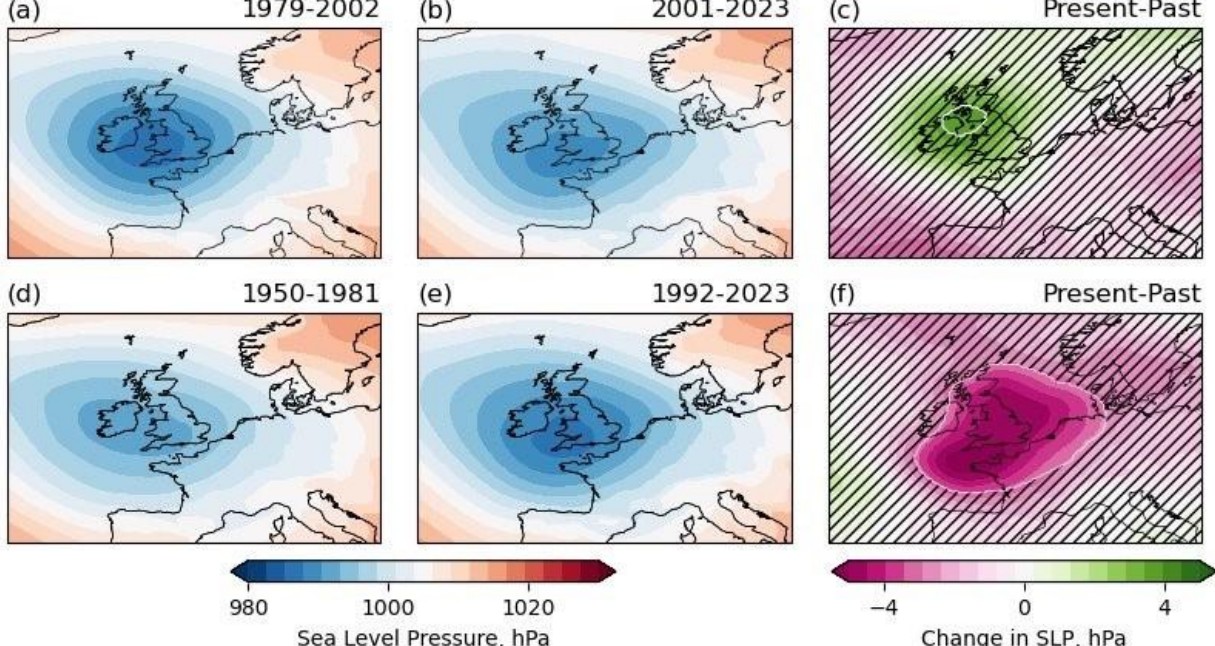

**Figure 2: Composites of analogues of Storm Babet from different time periods. (a-b)** The composites of the closest 1% analogue days from ERA-5 within the satellite period, 1979-2002 and 2001-2023. **(c)** The difference between the two periods, non-hashed regions indicate a statistically significant difference between the time periods. **(d-f)** as in (a-c), but for a longer time period, 1950-1981 and 1992-2023.

### 3.2 Multidecadal variability

Instead of comparing analogues from two time periods to assess the change through time, we can evaluate the variability through time. Using a measure of similarity, S (see Methods), we assess how the quality of analogues varies interannually from 1950 to 2022 (Fig.3). Higher values of S indicate the closest analogues for that year are more similar to the storm Babet.



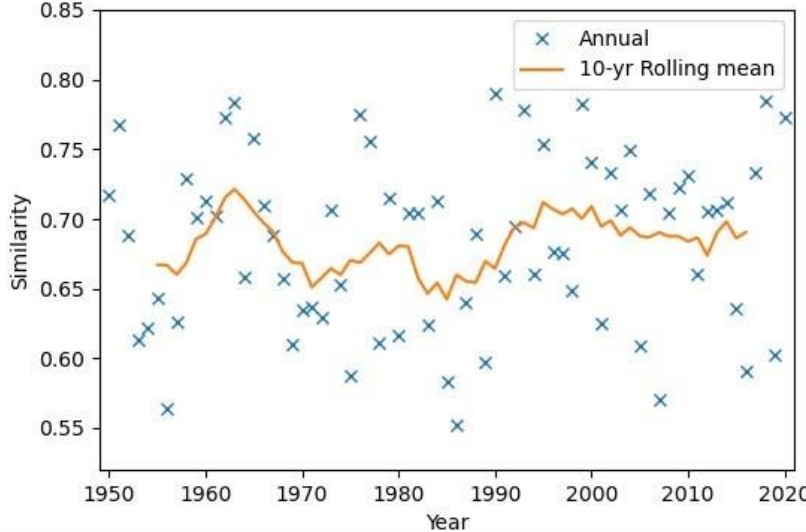

**Figure 3: Timeseries of Analogue Similarity.** The annual maximum of the September-October-November daily similarity values, Sx, to Storm Babet, 20th October 2023, for each year from 1950 to 2022 (blue crosses) and with a rolling 10-year mean applied (orange line).

We assess the timeseries of maximum annual (SON) analogue similarity, Sx (Fig.3). We find multidecadal variability, explaining the differing patterns for differing timeslices shown in Fig.2. When comparing the top analogues from two different time periods, we may be sampling two different phases of long-term variability, rather than assessing solely change caused by anthropogenic climate change.

As a side note, we did also calculate the Sx timeseries using 500 hPa geopotential height to assess analogues, as well as sea level pressure. We found the two Sx timeseries were highly correlated (Pearson's correlation score of .78, p-val = 1.2e-13), therefore similar results for the drivers of the variability would be found whichever variable is used.

AMV (Fig.4a) is known to influence the UK climate, and thus is likely a driver of the variability in Sx. Other drivers likely also play a role, such as the effects of climate change. We use multiple regression to separate the potential relative roles of decadal variability AMV and GMST in forcing the decadal variability in Sx. We find coefficients of 0.074 /°C for AMV and 0.003 /°C for GMST. AMV varies by ~0.5 °C over the past century, whereas GMST by ~1 °C - thus the GMST covariate has twice the impact, but even so, AMV dominates the variability in Sx. A significant, strong, correlation between AMV and Sx on a decadal timescale is shown (Pearson's correlation score of .53, p-val = 9.9e-6) (Fig.4b). When AMV is in a positive phase it appears that similar events are more likely. Assessing the correlation of Sx with GMST we do find a lower correlation than for AMV (Pearson's correlation score of .27, p-val = 0.04). We acknowledge that other drivers likely play a role, further work evaluating multiple drivers would be valuable.





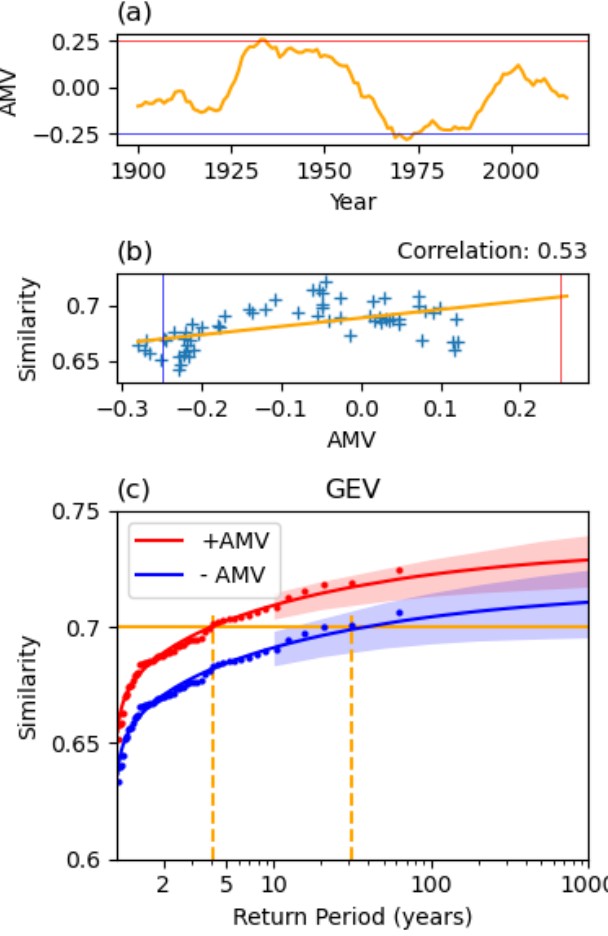

**Figure 4: Influence of AMV phase on Storm Babet-like storms. (a)** Timeseries of 10-year rolling mean of 1900-2017 AMV (September-October-November SST 25-60 °N, 7-75 °W minus regression on global mean temperature, as in van Oldenborgh et al 2009, based on HadSST 4.0.1.0, AMV [K]) in orange, with red horizontal lines indicating the values used for +AMV (+.25 °C) and –AMV (-.25 °C) phases. **(b)** 10-year rolling mean of Sx plotted against AMV for 1955-2017, with a best fit line (orange) and vertical lines indicating the AMV phases, as in **(a)**. **(c)** Return period of Sx with values adjusted to +AMV (red) and -AMV (blue). The solid orange line indicates S=0.7, with dashed vertical lines showing the return period of S=0.7 for +AMV (4 years) and –AMV (31 years).

### 3.3 Impact of different AMV phases

The shift in return period of Sx with AMV phase is assessed (Fig.4c). We show a significant shift between phases, when AMV is in a positive phase, events similar to Storm Babet are more likely – the chance of Sx>0.7 is 7.5 times more likely during AMV positive than negative (return period of 4 years in +AMV and 31 years in –AMV). Such events are likely to lead to wetter and windier conditions, which we can quantify (Fig.1, Fig.5). For rainfall, we assess North East Scotland (Fig.1e), for days where S>0.7 the mean rainfall is 3x greater, 8.4 mm/day compared to 2.8 mm/day for all days (Fig.5a). The North Sea is assessed for wind (Fig.1f), days with S>0.7 are 1.2x windier with a mean of 10.4 m/s compared to 8.5 m/s




for all days (Fig.5b). In both cases the differences between distributions are statistically significant using the Mann Whitney
test (p=2e-12 for rainfall and p=1e-7 for wind).

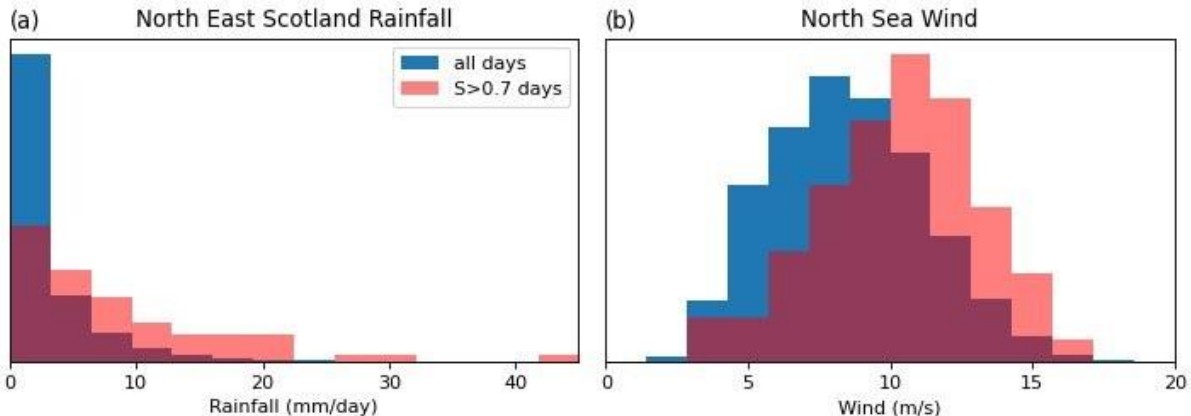

**Figure 5: Higher chance of surface impacts in days with greater similarity. (a)** Histograms of daily rainfall totals (mm/day) over North
East Scotland, 55.5-57.5N, 2-4W (shown on Fig.1e), the blue distribution indicates the values for every SON day, and red for days where
S>0.7. **(b)** As in (a) but for the average daily wind speed (m/s) over the North Sea, 55-59N, 0-8W (shown in Fig.1f).

## 4. Conclusions

Using analogue event sets to assess changes in extreme events is a growing area of research, and its use for event attribution
is rapidly developing (e.g. Climameter). Useful statements about changes in event frequency, intensity, and spatial
characteristics can be drawn from analogues sets (Thompson et al., in review). We show care must be taken when
interpreting the causes of differences between analogues within specified time periods – shifting the time periods could
significantly alter the findings. We present an alternative method of assessing changes in similarity through time which does
not assume a single, linear, trend. This allows the relative contributions of different factors, such as anthropogenic warming
and modes of internal climate variability, to be considered. For Storm Babet, we show that similar events are more likely
during positive AMV phase.

195         In recent years there has been a trend towards warmer temperatures in the North Atlantic, with unprecedented
temperatures observed in 2023 (Copernicus, 2024). The results suggest that if the current trends of amplified warming in
North Atlantic sea surface temperatures continue, we should expect to see more events similar to Babet.

        There is scope to develop this method for use in operational attribution.  We present an assessment of the link
between AMV phase and events similar to Storm Babet, but this method could be applied to other extreme weather events
with different modes of climate variability. For example, North American storms may vary with phase of the Pacific Decadal
Oscillation (Bond and Harrison, 2000). Multiple modes of variability may be considered at once, using multiple regression,
but this would increase uncertainty in the results.



Though we use only reanalysis data, the relationship between events similar to Babet and the AMV could be further tested using climate models, where temporal variability in the AMV can differ from that in the real world. Larger datasets, as can be provided by large ensembles of climate models, would allow the apparent influence of AMV phase to be explored further – and could also investigate other extreme weather events and climatic variability indices.

Attribution of surface impacts, such as the flooding caused by extreme rainfall, is a growing area of research (Scussolini et al., 2023). We find that analogues identified from atmospheric circulation can have consistent rainfall patterns, but a range of rainfall totals can occur with similar atmospheric patterns – as shown by the range of the S>0.7 day distributions in Fig.5. In the case of Storm Babet extreme rain impacted central England – which does not occur in the analogues of the event. Sea level pressure patterns alone do not determine the location of extreme rainfall, adding further complexity to impact attribution.

In this paper we consider that AMV and GMST can separately influence the likelihood of events similar to Storm Babet. We do not consider the drivers of the AMV – there is evidence that AMV is caused by a combination of other forcings, particularly sulphate aerosols, rather than intrinsic internal variability (Mann et al. 2021). We acknowledge that anthropogenic forcings may be influencing AMV, but this will not impact our key findings. If applying the method to different events and drivers it is important to consider the relationships between the drivers.

In conclusion, we present a novel method for assessing the relative roles of internal variability on specific atmospheric circulation patterns, by using atmospheric analogues. We find that events similar to Storm Babet are more likely to occur in positive AMV phase. With further development, this method could be used in event attribution to determine the relative importance of anthropogenic forcings and other drivers.

*Code and data availability.* The ERA5 reanalysis data are publicly available at https://cds.climate.copernicus.eu/. Atlantic Multidecadal Variability and Global Mean Surface Temperature timeseries are available from Climate Explorer (https://climexp.knmi.nl/). The python scripts used for analysis are available on github and zenodo (to be added on paper acceptance).

*Author contributions.* VT conceptualised the study, performed the data analysis, and produced the figures. All authors contributed to the discussion of the results and writing of the manuscript.

*Competing interests.* The authors declare that they have no known competing financial interests or personal relationships that could have appeared to influence the work reported in this paper.





*Acknowledgements.* This research has been supported by the KNMI multi-year strategic research funding (grant name MSO-ExtremeWeather). This study was partly supported by the European Union's Horizon 2020 research and innovation programme under grant agreement No 101003469 (XAIDA project).

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

# Appendix A

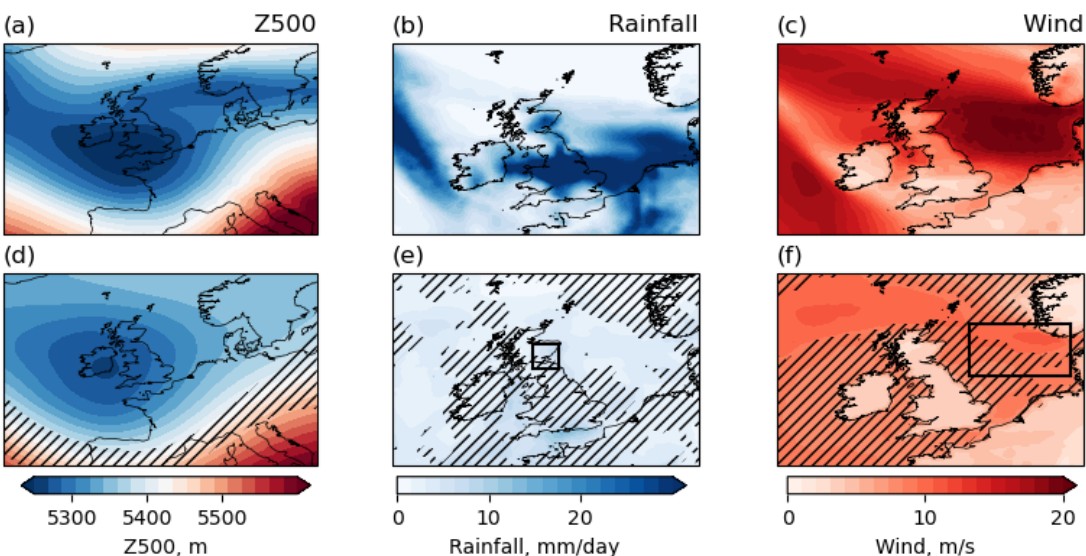

**Figure A1: Analogues of Storm Babet 500hPa geopotential height (Z500).** Similar to Fig.1, but using Z500 to identify analogues rather than sea level pressure. **(a-c)** The meteorological situation on 20th October 2023, showing **(a)** Z500 (m), **(b)** daily rainfall (mm/day), and **(c)** daily mean wind field (m/s). **(d-f)** Composites of the 20 closest analogues from 1950 to 2022, based on Z500, for **(d)** Z500 (), **(e)** daily rainfall (mm/day), and **(f)** daily mean wind field (m/s). All data from ERA-5. In (d-f) hashing indicates regions where the pattern is insignificant (mean anomaly < standard deviation). The regions indicated in **(e)**, North East Scotland, and **(f)**, North Sea, are assessed
further in Fig.5.





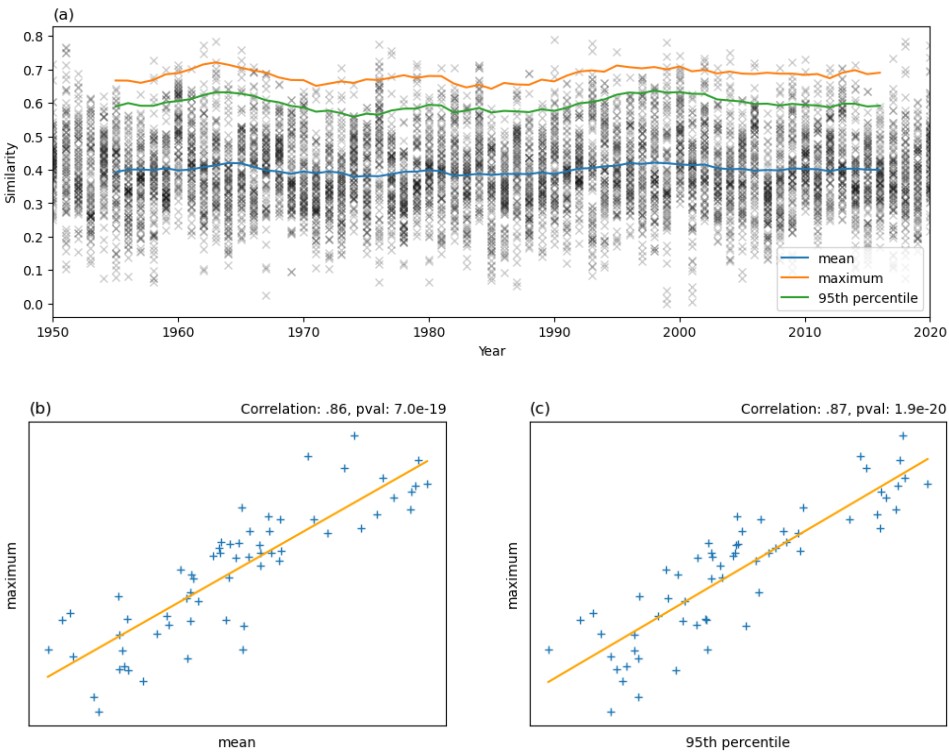


**Figure A2: Choice of similarity metric. (a)** Similarity values for every SON day each year, 1950-2020 (black crosses) with the timeseries of 10 year rolling mean of mean (blue), maximum (orange) and 95th percentile (orange). **(b)** Relationship between the 10 year rolling mean of annual maximum and mean similarity values, showing a Pearson's correlation of 0.86. **(c)** Relationship between the 10

year rolling mean of annual maximum and 95th percentile similarity values, showing a Pearson's correlation of 0.87.