# Peer review of "The influence of the Atlantic Multidecadal Variability on Storm Babet-like events"

_EGUsphere, 2024_

## Referee Comment (RC1)

**Review of "The influence of the Atlantic Multidecadal Variability on Storm Babet-like events"**

The work of Thompson et al. presents an investigation of the link between Atlantic Multidecadal Variability (AMV) and analogues of storm Babet (October 2023). The authors seek to investigate the sensibility of analogues averaging inside time slices as done by Faranda et al. (2022) to conclude on the influence of anthropogenic climate change on similar storms. The goal of the paper is to quantify the impact of multidecadal variability (AMV in particular) in the North Atlantic on the analogues found using multilinear regression on variability indices.

The paper is well-written and the research question is clearly stated. The latter is an important avenue of research for quantifying the sensibility of the analogues method to the long-term variability of the climate system and understand better what kind of conclusions can be drawn from this method as it is now used for attribution purposes (the authors cite the Climameter tool). Moreover, the work of Thompson et al. is a contribution to the literature on the dynamical evolution of the atmosphere under anthropogenic and natural forcings. This makes this work suited for publication in Weather and Climate Dynamics.

However, at this stage I recommend major revisions before accepting the paper for publication. In general, the paper lacks sensitivity tests to convince the reader about the robustness of the results found. But more importantly, the authors seem to claim to have found a causal link between AMV and analogues of storm Babet. I am not convinced that the arguments of the authors are sufficient to defend such a statement. Statements in the conclusion such as L193-194: "For Storm Babet, we show that similar events are more likely during positive AMV phase." and L196: "The results suggest that if the current trends of amplified warming in North Atlantic sea surface temperatures continue, we should expect to see more events similar to Babet." are not, or very weakly, supported by the analysis provided here. Moreover, if such statements were to be supported more strongly in a revised version of the manuscript, they should be compared to the scientific literature on this subject. See below for the details of my comments.

**Major comments:**

1. Sensitivity tests:
   a. The authors decided to use an arbitrary number of analogues in each period. While the number of analogues considered will always be more or less arbitrary, I think the authors should investigate the sensibility of their results to the number of analogues chosen (especially for Fig 1 and 2).
   b. Have you assessed the sensibility of your results to the spatial domain chosen for defining the analogues?
   c. In Fig 1 and Fig 2, how the statistical significance is obtained is not clear. The legend in Fig 1 indicates that it is obtained when the mean anomaly is below the standard deviation. I am not sure to understand what that means and I do not think this is a proper statistical test or procedure.
   d. L155-164: the results of the regression are not reported properly: there is no indication on the statistical significance of the regression, nor on the $R^2$, nor on the uncertainties and p values for the coefficients for AMV and GMST.

e. Fig 4c: I guess the shadings of the GEV represent the uncertainties. How were they obtained? Why do they begin only for return periods greater than 10 years?

f. Fig 4c: The authors should be more explicit in what they fit here. I guess that the location parameter depends linearly on AMV, but does it also depend on GMST? Please explain more clearly what GEV is fitted and with which fitting method.

g. L173-179: None of the values reported here have a confidence interval associated.

h. Fig 5:

    i. It is not clear to me why these two boxes were chosen. How much does this result depend on the region selected? You could rather show a map with the difference in the mean at each grid point to see how this pattern varies geographically.

    ii. I am not sure I understand the argument behind this figure. Using days for which S>0.7 is not completely equivalent to using days in positive vs negative phases of the AMV. Maybe you could find analogues in the two phases of the AMV and see the difference? This would be similar to what Cadiou et al. (2023) did for ENSO.

2. Impacts of AMV and causality statements:

    a. The authors use multilinear regression to assess the combined effect of AMV and GMST on Sx, a measure of the quality of the yearly best analogue of storm Babet. I will assume here that using Sx is correct (see below for my comment on this point) for estimating the sensibility of analogues. However, I am not sure how we should interpret the results of this regression. The authors seem to say that a significant AMV coefficient is sufficient to conclude to a causal effect. It may be true but the authors do not give enough arguments to support this claim. First, it is clear to me physically that if one finds a link between AMV and Sx, it is probably because AMV influences Sx rather than the contrary. The authors should explain that more clearly because it is currently implicit in their formulation. Second, to correctly estimate the impact of AMV on Sx, one needs to control for **all confounders**, i.e. for variables that have a causal impact on both AMV and Sx, and **only for these confounders**. The authors say in L 215: "We acknowledge that anthropogenic forcings may be influencing AMV, but this will not impact our key findings." Actually yes it would: any variable that would influence both AMV and Sx (such as anthropogenic forcings) and that is not taken into account in your regression will bias the estimation of the AMV coefficient. On the other hand, here the authors seem to suggest that GMST is such a confounder, which does not seem correct to me. Moreover the authors do not detail and discard other potential confounders (anthropogenic aerosols for example?). I recommend the work of Kretschmer et al. (2021) for clarifying these points.

    b. The correlation between AMV and Sx is rather weak (around 0.5 which means that only 25% of the variance is explained). Moreover, Fig 4b is quite worrying: it does not convince me that there is a linear relationship between these two quantities. If anything, it would suggest a second order relationship

(ie proportional to AMV**2). Finally, the authors project their results for a value of +0.25 for AMV which is far above what is ever observed in the data set. It seems to me quite problematic: how can you be sure that other data points will not have lower values of the similarity for high values of AMV as the negative trend for points on the right of AMV=0 suggests?

c. The key results of this work are based on the analysis of the yearly max Sx. I think this is problematic for several reasons.

    i. I do not see the point of using yearly maxima here. It is not clear how these yearly maxima reflect correctly the analogues found: how many of these maxima correspond to analogues used in Fig 1 and Fig 2? How many analogues do not correspond to yearly maxima? (there could be several analogues in a single year and none in other years).

    ii. The core of the results lies on the non-stationary GEV fitted on the Sx. Although this method is interesting, I am not sure we can interpret the results as straightforwardly as assumed by the authors. What the GEV gives is the probability that the yearly maximum of S is above a given threshold. It does not say how many analogues there would be per year and whether they will be good: it only gives a probability to the best one being very close. The authors say for example in L174: "events similar to Storm Babet are more likely – the chance of Sx>0.7 is 7.5 times more likely during AMV positive than negative", I am not sure this assertion is supported by the fit on Fig 4c because I do not think the fit of the GEV on the Sx can be straightforwardly linked to neither the quantity nor the quality of the analogues in the positive and negative phases of AMV.

    iii. It seems to me that for what the authors want to achieve, the GPD approach for the extremes of S is more suited because it takes into account the quantity of analogues in addition to the shape of the tail. I want to highlight the fact that using a GPD (or GEV) approach on the distance of the analogues of a point (ie linking extreme value theory and recurrence in dynamical systems) is a mathematical domain that has been well developed in the recent years (see Lucarini et al. 2016 for an exhaustive review) and that it is the mathematical foundation for the local dimension measure provided in Faranda et al. (2022).

**Minor comments:**

1. L27: typo "stormtims"
2. L95: maybe the number of the section is missing
3. L104: this equation should have a random term epsilon
4. L179: using the Mann-Whitney test to find differences in the distributions is okay but here you report differences in the averages of these distributions. A Welch's t-test seems more appropriate to me.
5. Figure A2: the color of the 95th percentile is indicated to be orange but is green in the figure. Also, why are there no units on the x and y axis of panels b and c ?

**References:**

Kretschmer, M., Adams, S. V., Arribas, A., Prudden, R., Robinson, N., Saggioro, E., & Shepherd, T. G. (2021). Quantifying causal pathways of teleconnections. *Bulletin of the American Meteorological Society*, 1-34.

Lucarini, V., Faranda, D., de Freitas, J. M. M., Holland, M., Kuna, T., Nicol, M., ... & Vaienti, S. (2016). *Extremes and recurrence in dynamical systems*. John Wiley & Sons.

Cadiou, C., Noyelle, R., Malhomme, N., & Faranda, D. (2023). Challenges in attributing the 2022 australian rain bomb to climate change. *Asia-Pacific Journal of Atmospheric Sciences*, *59*(1), 83-94.

---

## Author Comment (AC1)

**The influence of the Atlantic Multidecadal Variability on Storm Babet-like events**

Response to reviewer 2:

The draft focuses on storm Babet and its analogs in terms of sea-level-pressure patterns. Storm Babet occurred in October 2023, bringing significant rainfall and wind speeds that affected many part of the British Isles. Analogs of storm Babet are calculated from October to November and from 1950 to 2023. It is found that the pattern of such analogs has a time evolution consistent with the influence of the Atlantic Multidecadal Variability (AMV). A warm Atlantic corresponds to a more frequent occurrence of daily sea-level-pressure patterns similar to storm Babet.

The draft is concise but remains somewhat superficial regarding methods. It lacks discussion and analysis on the role of AMV. Additionally, a discussion about the existing literature and the limitations of the analyses conducted is missing. I recommend some major revisions.

We thank the reviewer for the detailed comments. We have revised the manuscript based on the comments, and those of another reviewer, and hope the reviewer finds it improved.

We have made many changes, including further statistical testing to show the robustness of the results, and a more detailed comparison with other scientific literature on the AMV and European storms. Point-by-point responses follow, with changes to the manuscript highlighted.

Major comments

1. The definition of the AMV (region used, smoothing applied or not) is lacking. There is a legend in Fig. 4 specifying that what is called the AMV is, in fact, the SST averaged in 25°N-60°N in September-October-November. However, what is usually called the AMV is a yearly SST average from 0° to 60°N, including the tropics from 0°N to 20°N. The dataset used to calculate the AMV is not clear (HadISST at L68, but HadSST 4.0.1.0 mentioned at L168). The smoothing applied and the dataset used for the GMST is lacking. Lastly, I suggest that the authors remove the external forcing using other methods, as the linear regression used might induce spurious connections with the Indo-Pacific (Deser and Philips, 2023). This is relevant as ENSO can be a

driver of fall European climate, which is excluded in the present draft (King et al., 2018).

There are multiple definitions of AMV in use. The definition of AMV used in this study is taken from van Oldenborgh et al (2009). By calculating AMV without including the tropics we remove some of the influence of ENSO and the Indo-Pacific. This is the version of AMV used on KNMI Climate Explorer. The dataset used is HadISST, this has been corrected in the text in the caption of Fig.4.

We used September-October-November (SON) to match the period used for analogues. Comparing the AMV timeseries for SON with the annual timeseries we find the two are highly correlated (shown below), thus changing to annual would make little difference to results. The method we have chosen allows the method to be applied to other events and other climatic indicators in a consistent manner - always taking the three months surrounding the event rather than an annual timeseries where sometimes an event could be at the start, and sometimes the end, of the period.

[Figure]

We have added further details of the GMST and AMV datasets used to the methods:

We use timeseries of AMV and global mean surface temperature (GMST) derived from HadISST data (Kennedy et al., 2019; Morice et al., 2021) retrieved from the KNMI Climate Explorer. The AMV index used takes only September-October-November SSTs, from 25-60°N, 7-75 °W, and is detrended by regressing against GMST (van Oldenborgh et al., 2009, Climate Explorer).- This is done to remove the effect of climate change in the AMV. By excluding the tropics the influence of ENSO and the Indo-Pacific on AMV is reduced (van Oldenborgh et al., 2009; Deser and Philips, 2023).

For all datasets a 10-year rolling mean is applied before regressing:

As AMV is a multidecadal mode of variability, a 10-year rolling mean is applied to the Sx, AMV, and GMST timeseries.

We take a smoothed GMST as a proxy for the external forcings, without removing any internal variability influences on GMST. This is common practice, as in WWA event attribution studies.

2. The links between the analogs and the AMV is not clear from the results shown. The links are presently deduced from correlations discussed at lines 155-163. However, the tests applied to deduce the p_value are not presented. The correlation of 0.53 given at line 60 is associated with a p-value of almost zero. However, given the few degrees of freedom (roughly 5, considering that there are 7 independent data points from the 7 decades), one can expect such a correlation is not significant at the 5% level. The physical process is also missing in the discussion. What is the process explaining that a warm Atlantic leads to more significant storms? This should be discussed more carefully, as such a link can be a statistical artifact.

Further statistical testing and the inclusion of p values has been added throughout the manuscript, including in the multiple regression results.

In section 3.2:

We use multiple regression to separate the potential relative roles of decadal variability AMV and GMST in forcing the decadal variability in Sx. The multiple regression model has an $R^2$ value of 0.45 (uncentred), with a p-value=1.3e-8, indicating a statistically significant relationship between similarity and the predictor

measures, AMV and GMST. We find coefficients of 0.074 /°C (p-value = 1.3e-4) for AMV and 0.003 /°C (p-value = 2.0e-8) for GMST. AMV varies by ~0.5 °C over the past century, whereas GMST by ~1 °C - thus the GMST covariate has twice the impact, but even so, AMV dominates the variability in Sx.

In response to your next comment, we now include a map of correlation between the analogue similarity timeseries (Sx) and SST globally, on an interannual timescale. This shows significant correlation in the North Atlantic on interannual timescales, as well as the decadal scale included in the text.

We have also added more about the physical reasons we would expect causality between AMV and analogues, particularly in the introduction (exact changes highlighted in the response to your minor comments).

3. L162-163 : the other potential driver of the change in analogs are not investigated. Perhaps a regression of the sea surface temperature on the Similarity time series would help show which region is the most important: the Pacific Ocean, the subtropical Atlantic or the subpolar Atlantic. The choice of looking at the AMV time series only seems otherwise arbitrary.

In the supplementary material we now include a map of the correlation of the analogue similarity timeseries (Sx) and SST for both an annual and 10-year rolling mean timeseries. This shows statistically significant correlation in the North Atlantic, justifying our choice of assessing the AMV timeseries. This is discussed in the methods:

We assess the relationship between Sx and AMV, AMV is chosen as literature supports a physical link between it and extratropical storms, as discussed in the introduction. We tested this by assessing the correlation between Sx and SSTs, showing statistically significant correlation over the North Atlantic on both an annual and 10-year rolling timescale (Fig.A3).

[Figure]

**Figure A4: Correlation between similarity and SSTs.** **(a)** Global map of the Pearson's correlation score between annual SST timeseries and the annual maximum similarity (Sx) timeseries, hashing indicates regions where the p-value is greater than 0.05, **(b)** As in (a) with a 10-year rolling mean applied to both the SST and Sx timeseries.

We do not assess other drivers, but in the discussion section we state that the method could be extended to include other modes of internal variability or drivers of climate change – which could be particularly relevant for other event types and locations.

4. The method of adjusting a GEV with the AMV is not explained at all. Please expand the paragraph L107-112. Did the authors try to include the GMST into the estimations of the return period as well?

We have adjusted the paragraph to clarify the method used. We do not adjust for GMST, only for AMV. Rather than assessing a factual world and counterfactual world we instead compare a '+AMV-only world' to a '-AMV-only world'.

Methods:

We find AMV is affecting Sx (see Results), and use extreme value theory to assess the change in return period of Sx with different AMV phase. We use the generalised extreme value distribution, as we are using annual (block) maxima (Coles, 2001; Philip et al., 2020). We assume that the trend shifts with AMV, this is factored into the GEV by allowing the location parameter to be linearly dependent on the AMV. The distribution is then evaluated at two values of AMV: +0.25 °C (positive phase) and –0.25 °C (negative phase). We make no adjustment for changes in GMST, only AMV. This is similar to how GEV can be used to assess a factual and counterfactual world, e.g. with and without climate change, but in our case we use only AMV as a covariate (van Oldenburgh et al., 2021; Philip et al., 2020).

Minor comments:

L23-25: Can the authors describe more the previous methods and results that led to the conclusion that the AMV or PDO may have an influence on events like storm Babet?

The following has been added to the text:

Their method identified analogues in two different timeslices, for Storm Babet they found these analogues occurred in statistically different phases of AMV and PDO in the past and present.

L30-32: The other potential drivers of the AMV need to be discussed as it is a controversial topic. The atmospheric forcing plays a role. The external forcing also explains a large part of the AMV (Klavans et al., 2022).

We have expanded the text to include details of evidence for external radiative forcings, and added further references:

External radiative forcings may also influence the phasing and magnitude of the AMV. For example, Undorf et al., (2018) and Watanabe and Tatebe (2019) find greenhouse gases and aerosols play a key role in the timing, yet prior to 1950 volcanic eruptions coincide with phase changes (Birkel et al., 2018). With changes to the external radiative forcings on the climate system the drivers of AMV may also be changing, Klavens et al., (2022) find a growing contribution from external drivers.

L34-36: Provide more details on how the AMV affects the NAO. The references given all presents different mechanism that can be further presented to the reader.

We have explained the mechanisms further:

The AMV can be linked to European storminess through its influence on the wintertime North Atlantic Oscillation (NAO) (Ruggieri et al., 2021; Msadek et al. 2011; Davini et al. 2015; Peings and Magnusdottir 2014, 2016). The AMV influences the spatial pattern of the NAO, influencing storm tracks across the North Atlantic (Börgel et al.,

2020). Climate model experiments nudged to specified AMV states are often used to investigate the mechanisms. For example. Ruggieri et al (2021) showed that the low-level jet differs significantly between AMV states, with a positive phase leading to a more southerly jet in the eastern Atlantic. This agrees Peings and Magnusdottir (2016), who showed that temperature changes in the extratropics are required to force a shift in the NAO and storm tracks. Msadek et al. (2011) showed that, alongside the southward shift in the stormtrack there is an intensification of the subtropical jet during positive AMV.

L36-37: Provide more details on the influence of AMV on the stormtrack, and its link with the NAO. See also Varino et al. (2019)

Further details have been added:

These mechanisms, found in climate models, are also supported by reanalysis data which shows a stronger relationship between AMV and North Atlantic extratropical storms frequency, than with the accelerated polar warming seen in recent decades (Valino et al., 2019).

L42-43: "it is often inferred that the change in analogues is the effect of climate change" Please provide references.

An example has been provided:

For example, Faranda et al. (2024) test for significantly different AMO, PDO, and ENSO states between analogues phases, and in the absence of these assume the change is climate change driven.

L60: Can the authors describe Figure A1 and explain which aspects of the SLP-analogs are better than the 500 hPa-analogs.

The following description has been added:

Analogue composite identified using 500 hPa geopotential height does not show rainfall over the regions impacted by Storm Babet. The analogues do show the same wind signal as the sea level pressure analogues, but it is much weaker – and thus further from the observed event.

L75: Is the Euclidian distance the root mean squared error of the fields using area weighting?

Good point – it should be, but was not. We now apply an area weighting to the fields before calculating the Euclidean distance. For this event definition it did not change the analogues identified, but for other events this may not be the case. The scripts (written in python using the iris library) will be made available on publication, and a description of this is has been added to the Methods sections:

We determine similarity by calculating the area-weighted Euclidean distance of the sea level pressure field over the event domain (Faranda et al., 2022).

Fig 1 : the rainfall is strongly different in Fig. 1b and 1e. Was it expected? Can an analog built using only sea-level-pressure expected to capture high-precipitation events? Maybe part of the key dynamics of the event is missing in the analog, such an atmospheric rivers. Maybe the precipitation is not well captured in ERA5. Did the authors try other precipitation observations?

Other precipitation observations agree with the ERA5 precipitation. For example, the UK Met Office reports the daily rainfall totals for 20th October as:

[Figure]

It was surprising that the England / Wales rainfall is not captured by the analogues – only the Scottish rainfall. Looking into it we found that the report from Climameter on the event found similar (ClimaMeter - 2023/10/21 Storms Babet Aline). We have also been working with other groups who also find the rainfall over England and Wales is not captured so well in initialised forecast ensembles. With these groups we are carrying out further work investigating these differences and potential changes in a future climate. Atmospheric rivers are being investigated in that work – but as of yet we have no definite findings.

Interestingly, in the new Fig.5 we show that the top 1% of analogues in AMV+ years only do show some of the England/Wales rainfall pattern. We have added a comment about this in section 3.3:

For rainfall the AMV+ analogues show greater similarity to the observed event over England and Wales – a region where the analogues over the whole period failed to capture the event (Fig.1b/e).

Fig. 1: please explain in the methods the test implemented to show statistical significance. Only comparing the composite to the standard deviation of the field is too arbitrary.

The statistical testing has now been changed to instead use a one-sided t-test, with hashing indicating regions with p>0.05 (insignificant). The figure and caption have been updated.

Methods text has been updated:

We perform significance testing on the composites. When investigating analogues from one timeslice we assess where the signal in the composite of the analogues is statistically different to zero, using a one-sided t-test (p<0.05) (Fig.1).

Fig. 2: the test used for the statistical significance is not presented or explained.

We have updated the figure caption, and added the following to the methods:

When comparing the analogues from different time periods we apply two-sided t-tests to each gridpoint, assuming equal variance, if p<0.05 the distributions are assumed to differ significantly (Fig.2).

L154: I am not convinced that the Sx time series based on 500-hPa would have similar results as it is correlated with the Sx time series based on sea-level-pressure. The authors should apply their analyses to the two time series (i.e. Sx based on sea-level-pressure and Sx based on 500-hPa geopotential).

We had not included applying the analysis to 500 hPa geopotential height – as in Fig.A1 we show analogues found using Z500 did not agree well with the observed event, in terms of rain or wind.

Despite of this, we have carried out the analysis – finding strong correlation between the time series of similarity from the two variables, shown below:

[Figure]

L163: "other drivers likely play a role" I do not understand why the authors did not investigate maps of SST anomalies associated with their time series. The choice of only investigating the AMV looks like cherry-picking.

See our response to major comment 3, we have now added a map as suggested and this shows the North Atlantic has greatest correlation with the analogue similarity timeseries.

L175-180 : the analysis shown in Fig. 5 is not really about the impacts of different AMV phases (see name of section 3.3); it is about the impacts of the similarity variations.

We have now changed Fig.5 to instead show maps of the top analogues in +AMO compared to –AMO, the text has also been changed:

[Figure]

Text in both the methods and in section 3.3 has been changed to describe the new figure:

Methods:

**2.4 Analogues in different AMV phases**

We assess the differences between the analogues in different AMV phases by splitting the years into three groups based on AMV. We include the lower third (AMV index below –0.14) as –AMV years and upper third (AMV index above 0.04) as +AMV years. For those years we identify the top 1 % of analogues, using the methods described in section 2.2. Composites of these analogues are calculated (Fig.5). The statistical significance is calculated using a 2-sided t-test between each gridpoint for the AMV+ and AMV- analogues, if p<0.05 the distributions are assumed to differ significantly.

Sec3.2:

We assess the difference between the top 1% of analogues for each AMV phase (Fig.5). For rainfall the AMV+ analogues show greater similarity to the observed event over England and Wales – a region where the analogues over the whole period failed to capture the event (Fig.1b/e). For wind, we also find the AMV+ analogues are more similar to the event than AMV- with statistically significant stronger winds in the southern North Sea during AMV+ years. In agreement with Fig.4, these results suggest events more similar to Storm Babet are more likely during AMV+ years. The spatial pattern of surface impacts – both for extreme wind and rain, are more similar to Storm Babet in AMV+ years (Fig.5).

L177: "is 3x greater" changed in text '3 times greater'

L178 : "1.2x windier" changed in text '1.2 times windier'

L188: "climameter" Can the authors explain this word?

Climameter refers to the rapid event analysis framework used on www.climameter.org. Led by Davide Faranda at IPSL, France, it produces rapid event studies.

In the text it has been replaced by two references:

> Faranda, D., Messori, G., Coppola, E., Alberti, T., Vrac, M., Pons, F., Yiou, P., Saint Lu, M., Hisi, A. N. S., Brockmann, P., Dafis, S., and Vautard, R.: ClimaMeter: Contextualising Extreme Weather in a Changing Climate, EGUsphere [preprint], https://doi.org/10.5194/egusphere-2023-2643, 2023.

> Ginesta, M., and Faranda, D.: Strong winds in storms Babet & Aline likely strengthened by both human-driven climate change and natural variability, ClimaMeter. (climameter.org), 2023.

L189: "Thompson et al., in review" Can the authors make this paper available?

This paper is now published, and has been updated in the draft:

Changing dynamics of Western European summertime cut-off lows: A case study of the July 2021 flood event - Thompson - Atmospheric Science Letters - Wiley Online Library

L201-202 : "Multiple modes of variability may be considered at once, using multiple regression, but this would increase uncertainty in the results" I do not understand why investigating the role of other modes of variability through the use of multiple regression would increase the uncertainty.

This sentence has been changed:

Using multiple regression, multiple modes of variability may be considered at once

Fig A2 (a). Can the author explain in the legend what the green and orange lines represent?

The caption has been corrected to state the 95$^{th}$ percentile is green and maximum orange (as in the figure legend).

References:

Deser, C., & Phillips, A. S. (2023). Spurious Indo-Pacific connections to internal Atlantic Multidecadal Variability introduced by the global temperature residual method. *Geophysical Research Letters*, 50, e2022GL100574. https://doi.org/10.1029/2022GL100574

King, M. P., Herceg-Bulić, I., Bladé, I., García-Serrano, J., Keenlyside, N., Kucharski, F., ... & Sobolowski, S. (2018). Importance of late fall ENSO teleconnection in the Euro-Atlantic sector. *Bulletin of the American Meteorological Society*, *99*(7), 1337-1343.

Klavans, J. M., A. C. Clement, M. A. Cane, and L. N. Murphy, 2022: The Evolving Role of External Forcing in North Atlantic SST Variability over the Last Millennium. *J. Climate*, **35**, 2741–2754, https://doi.org/10.1175/JCLI-D-21-0338.1.

Varino, F., Arbogast, P., Joly, B., Riviere, G., Fandeur, M. L., Bovy, H., & Granier, J. B. (2019). Northern Hemisphere extratropical winter cyclones variability over the 20th century derived from ERA-20C reanalysis. *Climate dynamics*, *52*, 1027-1048.

---

## Author Comment (AC2)

**The influence of the Atlantic Multidecadal Variability on Storm Babet-like events**

Response to reviewer 1:

Review of "The influence of the Atlantic Multidecadal Variability on Storm Babet-like events"

The work of Thompson et al. presents an investigation of the link between Atlantic Multidecadal Variability (AMV) and analogues of storm Babet (October 2023). The authors seek to investigate the sensibility of analogues averaging inside time slices as done by Faranda et al. (2022) to conclude on the influence of anthropogenic climate change on similar storms. The goal of the paper is to quantify the impact of multidecadal variability (AMV in particular) in the North Atlantic on the analogues found using multilinear regression on variability indices.

The paper is well-written and the research question is clearly stated. The latter is an important avenue of research for quantifying the sensibility of the analogues method to the long-term variability of the climate system and understand better what kind of conclusions can be drawn from this method as it is now used for attribution purposes (the authors cite the Climameter tool). Moreover, the work of Thompson et al. is a contribution to the literature on the dynamical evolution of the atmosphere under anthropogenic and natural forcings. This makes this work suited for publication in Weather and Climate Dynamics.

However, at this stage I recommend major revisions before accepting the paper for publication. In general, the paper lacks sensitivity tests to convince the reader about the robustness of the results found. But more importantly, the authors seem to claim to have found a causal link between AMV and analogues of storm Babet. I am not convinced that the arguments of the authors are sufficient to defend such a statement. Statements in the conclusion such as L193-194: "For Storm Babet, we show that similar events are more likely during positive AMV phase." and L196: "The results suggest that if the current trends of amplified warming in North Atlantic sea surface temperatures continue, we should expect to see more events similar to Babet." are not, or very weakly, supported by the analysis provided here. Moreover, if such statements were to be supported more strongly in a revised version of the manuscript, they should be compared to the scientific literature on this subject. See below for the details of my comments.

We thank the reviewer for the detailed comments, and are pleased we have conveyed the importance of the research. We have revised the manuscript based on the comments, and those of another reviewer, and hope the reviewer finds it improved.

We have made many changes, including further statistical testing to show the robustness of the results, and a more detailed comparison with other scientific literature on the AMV and European storms to help support the case for causality.

**Major comments:**

**1. Sensitivity tests:**

a. The authors decided to use an arbitrary number of analogues in each period. While the number of analogues considered will always be more or less arbitrary, I think the authors should investigate the sensibility of their results to the number of analogues chosen (especially for Fig 1 and 2).

Indeed a different number of analogues could be chosen, and for different types of events it could be that this number should be adjusted (depending how rare the event itself is). We have added further explanation of our choice in the methods, and added a figure showing how the quality of the analogues varies with different analogue numbers, based on Fig.1:

*Sec2.2:* *Using too many analogue days would lead to the weaker analogues being too different to the event day to give useful information, too few prevents meaningful assessment. Fig.A2 shows the impact of different amounts of analogues on the analogue composites. The decision to use 1% of days as analogues is based on earlier studies*

*(Faranda et al., 2022; Thompson et al., 2024; Faranda et al., 2024).*

[Figure]

**Figure A2: The influence of the number of analogues used. (a-c)** The meteorological situation on 20th October 2023, showing (a) sea level pressure (hPa), (b) daily rainfall (mm/day), and (c) daily mean wind field (m/s). **(d-f)** Composites of the 60 closest analogues from 1950 to 2022, based on sea level pressure, for (d) sea level pressure (hPa), (e) daily rainfall (mm/day), and (f) daily mean wind field (m/s). **(g-i)** As in (d-f) for the closest 20 analogues. **(j-l)** As in (d-f) for the closest 10 analogues. All data from ERA5. In (d-l) unhashed regions indicate where the signal in the composite of analogues is statistically different to zero, using a one-sided t-test ($p < 0.05$).

b. Have you assessed the sensibility of your results to the spatial domain chosen for defining the analogues?

As is the case in many climate event assessments – such as the World Weather Attribution rapid attribution studies and the ClimaMeter assessments - we do have to define a spatial domain over which we assess. In the case of this study the domain was chosen to include

the main dynamical features – i.e. the low pressure system. The domain we choose is 40 °N to 65 °N, 20 °W to 20 °E (as shown in Fig1a).

We did explore different domains, for example the region used in Ginesta & Faranda (2003) 30 °N to 60 °N, 20 °W to 20 °E. We found much overlap in event dates (50 % of events show perfect agreement and a further 25 % are likely from the same event, but a day or two out). That study chose a more southerly region to try to include Storm Aline, which impacted Spain at the same time as Storm Babet.

Using too large a region will result in less close analogues being identified. Too small a region may lead to analogues which do not include dynamical features key to the development of the events impacts (such as large scale rainfall). The choice of domain requires expert judgement to ensure the key dynamical features are included, but the domain is not so large that analogues identified are too unusual.

c. In Fig 1 and Fig 2, how the statistical significance is obtained is not clear. The legend in Fig 1 indicates that it is obtained when the mean anomaly is below the standard deviation. I am not sure to understand what that means and I do not think this is a proper statistical test or procedure.

We have changed the statistical significance testing. For figure 1 we assess where the signal in the composite of the analogues is statistically different to zero, using a one-sided t-test ($p<0.05$). For figure 2 we assess the difference between the analogue sets, the significance is calculated using a two-sided t-test ($p<0.05$). The methods, figures, and figure captions have been updated to reflect the change.

Sec2.2 changes:

*We perform significance testing on the composites. When investigating analogues from one time period we assess where the signal in the composite of the analogues is statistically different to zero, using a one-sided t-test ($p<0.05$) (Fig.1). When comparing the analogues from different time periods we apply two-sided t-tests to each gridpoint, assuming equal variance, if $p<0.05$ the distributions are assumed to differ significantly (Fig.2).*

d. L155-164: the results of the regression are not reported properly: there is no indication on the statistical significance of the regression, nor on the $R^2$, nor on the uncertainties and p values for the coefficients for AMV and GMST.

The statistical significance has been indicated by adding the p-values for each coefficient. We have also now included the $R^2$ and p value for the multiple regression, which indicate that there is a relationship between the similarity measure and the predictors (GMST and AMV).

Sec 3.2 changes:

*The multiple regression model has an R² value of 0.45 (uncentred), with a p-value=1.3e-8, indicating a statistically significant relationship between similarity and the predictor measures, AMV and GMST. We find coefficients of 0.074 /°C (p-value = 1.3e-4) for AMV and 0.003 /°C (p-value = 2.0e-8) for GMST.*

e. Fig 4c: I guess the shadings of the GEV represent the uncertainties. How were they obtained? Why do they begin only for return periods greater than 10 years?

Yes it is uncertainties, at 95% confidence intervals. The caption of the figure has been updated to clarify this. The uncertainty bands have been extended to cover the full range.

f. Fig 4c: The authors should be more explicit in what they fit here. I guess that the location parameter depends linearly on AMV, but does it also depend on GMST? Please explain more clearly what GEV is fitted and with which fitting method.

The method for adjusting the similarity data based on AMV is explained at the end of the methods section (line 114 onwards). We have expanded this to clarify that there is no adjustment made for changes in GMST, only AMV.

Sec 2.3 text:

*We find AMV is affecting Sx (see Results), and use extreme value theory to assess the change in return period of Sx with different AMV phase. We use the generalised extreme value distribution, as we are using annual (block) maxima (Coles, 2001; Philip et al., 2020). We assume that the trend shifts with AMV, this is factored into the GEV by allowing the location parameter to be linearly dependent on the AMV. The distribution is then evaluated at two values of AMV: +0.25 °C (positive phase) and –0.25 °C (negative phase). We make no adjustment for changes in GMST, only AMV. This is similar to how GEV can be used to assess a factual and counterfactual world, e.g. with and without climate change, but in our case we use only AMV as a covariate (van Oldenburgh et al., 2021; Philip et al., 2020).*

g. L173-179: None of the values reported here have a confidence interval associated.

We have added uncertainties at the 95% confidence level - as described in the methods section.

h. Fig 5:

    i. It is not clear to me why these two boxes were chosen. How much does this result depend on the region selected? You could rather show a map with the difference in the mean at each grid point to see how this pattern varies geographically.

        We have changed Fig.5 to maps following your next comment.

    ii. I am not sure I understand the argument behind this figure. Using days for which S>0.7 is not completely equivalent to using days in positive vs negative phases of the AMV. Maybe you could find analogues in the two phases of the AMV and see the difference? This would be similar to what Cadiou et al. (2023) did for ENSO.

Thank-you for pointing out Cadiou et al (2023) - a very relevant paper that I had not yet seen. Indeed, the method used by Cadiou et al makes more sense as we then capture the differences in impacts of the top analogues in each AMV state. Fig.5 has been changed to compare the top 1% of AMV+ and AMV- analogues (as now described in the methods), as shown below:

[Figure]

Text in both the methods and in section 3.3 has been changed to describe the new figure:

Methods:

**2.4 Analogues in different AMV phases**

We assess the differences between the analogues in different AMV phases by splitting the years into three groups based on AMV. We include the lower third (AMV index below –0.14) as –AMV years and upper third (AMV index above 0.04) as +AMV years. For those years we identify the top 1 % of analogues, using the methods described in section 2.2. Composites of these analogues are calculated (Fig.5). The statistical significance is calculated using a 2-sided t-test between each gridpoint for the AMV+ and AMV- analogues, if p<0.05 the distributions are assumed to differ significantly.

Sec3.2:

We assess the difference between the top 1% of analogues for each AMV phase (Fig.5). For rainfall the AMV+ analogues show greater similarity to the observed event over England and Wales – a  region where the analogues over the whole period failed to capture the event (Fig.1b/e). For wind, we also find the AMV+ analogues are more similar to the event than AMV- with statistically significant stronger winds in the southern North Sea during AMV+ years. In agreement with Fig.4, these results suggest events more similar to Storm Babet are more likely during AMV+ years. The spatial pattern of surface impacts – both for extreme wind and rain, are more similar to Storm Babet in AMV+ years (Fig.5).

**2. Impacts of AMV and causality statements:**

a. The authors use multilinear regression to assess the combined effect of AMV and GMST on Sx, a measure of the quality of the yearly best analogue of storm Babet. I will assume here that using Sx is correct (see below for my comment on this point) for estimating the sensibility of analogues. However, I am not sure how we should interpret the results of this regression. The authors seem to say that a significant AMV coefficient is sufficient to conclude to a causal effect. It may be true but the authors do not give enough arguments to support this claim. First, it is clear to me physically that if one finds a link between AMV and Sx, it is probably because AMV influences Sx rather than the contrary. The authors should explain that more clearly because it is currently implicit in their formulation.

In the introduction we have added further details on the probable mechanisms of a physical link between AMV and Sx, and why we assume AMV is influencing Sx and not vice versa.

Following the comments of another reviewer, we have added a further supplementary figure. This figure assesses correlation of Sx with SST globally, to support the choice of AMV for the multiple regression.

Additions to the introduction:

External radiative forcings may also influence the phasing and magnitude of the AMV. For example, Undorf et al., (2018) and Watanabe and Tatebe (2019) find greenhouse gases and aerosols play a key role in the timing, yet prior to 1950 volcanic eruptions coincide with phase changes (Birkel et al., 2018). With changes to the external radiative forcings on the climate system the drivers of AMV may also be changing. Klavens et al., (2022) find a growing contribution from external drivers.  ....  The AMV influences the spatial pattern of the NAO, influencing storm tracks across the North Atlantic (Börgel et al., 2020). Climate model experiments nudged to specified AMV states are often used to investigate the mechanisms. For example, Ruggieri et al (2021) showed that the low-level jet differs significantly between AMV states, with a positive phase leading to a more southerly jet in the eastern Atlantic. This agrees Peings and Magnusdottir (2016), who showed that temperature changes in the extratropics are required to force a shift in the NAO and storm tracks. Msadek et al. (2011) showed that, alongside the southward shift in the stormtrack there is an intensification of the subtropical jet during positive AMV. These mechanisms, found in climate models, are also supported by reanalysis data which shows a stronger relationship between AMV and North Atlantic extratropical storms frequency, than with the accelerated polar warming seen in recent decades (Valino et al., 2019).

 Second, to correctly estimate the impact of AMV on Sx, one needs to control for all confounders, i.e. for variables that have a causal impact on both AMV and Sx, and only for these confounders. The authors say in L 215: "We acknowledge that anthropogenic forcings may be influencing AMV, but this will not impact our key findings." Actually yes it would: any variable that would influence both AMV and Sx (such as anthropogenic

forcings) and that is not taken into account in your regression will bias the estimation of the AMV coefficient. On the other hand, here the authors seem to suggest that GMST is such a confounder, which does not seem correct to me. Moreover the authors do not detail and discard other potential confounders (anthropogenic aerosols for example?). I recommend the work of Kretschmer et al. (2021) for clarifying these points.

We have changed L215:

We acknowledge that anthropogenic forcings may be influencing AMV and we do use an AMV timeseries with GMST regressed out, but with only a short observational record available it is difficult to separate the drivers. If applying the method to different events and drivers it is important to consider the relationships between the drivers. Large ensembles of climate models would provide a testbed to investigate the conditional likelihood of events similar to Storm Babet under combinations of drivers, providing a range of states of internal variability.

We have also made changes to the methods text, and added more on causality and the physical links to the introduction (see comment above):

This is done to remove the effect of climate change in the AMV, and will ensure we do not identify spurious correlation between AMV and analogues driven by GMST (Kretschmer et al., 2021).

b. The correlation between AMV and Sx is rather weak (around 0.5 which means that only 25% of the variance is explained). Moreover, Fig 4b is quite worrying: it does not convince me that there is a linear relationship between these two quantities. If anything, it would suggest a second order relationship (ie proportional to AMV**2). Finally, the authors project their results for a value of +0.25 for AMV which is far above what is ever observed in the data set. It seems to me quite problematic: how can you be sure that other data points will not have lower values of the similarity for high values of AMV as the negative trend for points on the right of AMV=0 suggests?

We had chosen an AMV of +.25 as it is the highest in the observational record going back further than the Sx record. It does indeed make sense to reduce this maximum to the upper end of what is seen in the Sx record – thus we have altered this down to +0.1. The conclusions are unaltered, and the difference between +AMV and –AMV is increased due to the identification of an error in the script. The new GEV figure has been verified using the Climate Explorer tool shown here (1975 ~ -25K AMV, thus 1975+.35 ~ 0.1 K AMV):

[Figure]

Indeed, with few data points above +0.1K AMV it is difficult to ascertain the relationship between Sx and AMV. We now highlight this issue further in the text:

We note, with only a short observational time series there is still uncertainty in this relationship, ideally more years of +AMV would be included but this is not available using ERA5. Further testing of the relationship with other observational datasets and climate model data would be useful, as highlighted in the conclusions.

c. The key results of this work are based on the analysis of the yearly max Sx. I think this is problematic for several reasons.

  i. I do not see the point of using yearly maxima here. It is not clear how these yearly maxima reflect correctly the analogues found: how many of these maxima correspond to analogues used in Fig 1 and Fig 2? How many analogues do not correspond to yearly maxima? (there could be several analogues in a single year and none in other years).

  Using yearly maxima allows us to produce an annual timeseries, which we need to allow us to carry out the multiple regression against GMST and AMV. If we instead took the analogues used in Fig1 and Fig2, or some other definition of the top analogues, we would not get values for every year but instead clusters of analogues in certain periods (as shown by the clusters of high similarity events in Fig.A2a). To enable this method to be applied to other events an annual timeseries is required, to allow assessment of the contribution of various modes of variability.

  ii. The core of the results lies on the non-stationary GEV fitted on the Sx. Although this method is interesting, I am not sure we can interpret the results as straightforwardly as assumed by the authors. What the GEV gives is the probability that the yearly maximum of S is above a given threshold. It does not say how many analogues there would be per year and whether they will be good: it only gives a probability to the best one being very close. The authors say for example in L174: "events similar to Storm Babet are more likely – the chance of Sx>0.7 is 7.5 times more likely during AMV positive than negative", I am not sure this assertion is supported by the fit on Fig 4c because I do not think the fit of the GEV on

the Sx can be straightforwardly linked to neither the quantity nor the quality of the analogues in the positive and negative phases of AMV.

We have reworded parts of the interpretation to highlight that we are assessing the probability of the most similar event each year, rather than the similarity of each individual day:

We show a significant shift between phases, when AMV is in a positive phase, events similar to Storm Babet are more likely – the chance of the most similar event in a year showing Sx>0.7 is ~7.5 times more likely during AMV positive than negative

We argue that the fit of the GEV can be linked to the quality of the top analogue of a given year, though we agree it is not able to indicate anything about the quantity of analogues beyond the chance of none above a certain threshold. We use the annual maxima, rather than events above a threshold, to provide a timeseries which allows assessment against annual modes of climate variability.

iii. It seems to me that for what the authors want to achieve, the GPD approach for the extremes of S is more suited because it takes into account the quantity of analogues in addition to the shape of the tail. I want to highlight the fact that using a GPD (or GEV) approach on the distance of the analogues of a point (ie linking extreme value theory and recurrence in dynamical systems) is a mathematical domain that has been well developed in the recent years (see Lucarini et al. 2016 for an exhaustive review) and that it is the mathematical foundation for the local dimension measure provided in Faranda et al. (2022).

The GPD approach would perhaps be more appropriate for this case specifically – but the interannual similarity timeseries of the maximum event each year is required to assess the possible contribution of specific modes of variability or drivers of the change (in this case AMV and GMST). Without using an annual timeseries we would have been unable to use the multiple regression method used to investigate the possible relative roles of AMV and GMST in predicting the Sx variable. For consistency, throughout the study we use the Sx timeseries and therefore block maxima for the GEV.

**Minor comments:**

1. L27: typo "stormtims"

Corrected to 'storms'

2. L95: maybe the number of the section is missing

Indeed, I have added '2.3'

3. L104: this equation should have a random term epsilon

We have replaced the constant with a random noise term, which would also incorporate any constant:

Linear multiple regression is used to identify the potential relative influence of AMV and GMST on the chance of analogues occurring. Sx can be predicted using AMV and GMST through the equation:

$$Sx = \alpha\ AMV + \beta\ GMST \underline{+ \varepsilon}$$

Where $\alpha$ is the coefficient for AMV, $\beta$ is the coefficient for GMST, and $\varepsilon$ a random noise term. The coefficients can be used to interpret which has greater influence.

4. L179: using the Mann-Whitney test to find differences in the distributions is okay but here you report differences in the averages of these distributions. A Welch's t-test seems more appropriate to me.

As we have changed Fig.5 we no longer assess the differences between these distributions. In the new Fig.5 (and earlier figures comparing two analogue sets) we do use a Welch's t-test, as described in the methods and figure captions.

5. Figure A2: the color of the 95th percentile is indicated to be orange but is green in the figure. Also, why are there no units on the x and y axis of panels b and c ?

The caption has been corrected to state green for 95th percentile, and axis value labels have been added.

**References**:

Kretschmer, M., Adams, S. V., Arribas, A., Prudden, R., Robinson, N., Saggioro, E., & Shepherd, T. G. (2021). Quantifying causal pathways of teleconnections. Bulletin of the American Meteorological Society, 1-34.

Lucarini, V., Faranda, D., de Freitas, J. M. M., Holland, M., Kuna, T., Nicol, M., ... & Vaienti, S. (2016). Extremes and recurrence in dynamical systems. John Wiley & Sons.

Cadiou, C., Noyelle, R., Malhomme, N., & Faranda, D. (2023). Challenges in attributing the 2022 australian rain bomb to climate change. Asia-Pacific Journal of Atmospheric Sciences, 59(1), 83-94.